# AlignFix: Fixing Adversarial Perturbations by Agreement Checking for Adversarial Robustness against Black-box Attacks

**Ashutosh Kumar Nirala**                                  *aknirala@iastate.edu*
*Department of Computer Science*
*Iowa State University*

**Jin Tian**                                               *jin.tian@mbzuai.ac.ae*
*Professor of Machine Learning*
*Mohamed bin Zayed University of Artificial Intelligence*

**Olukorede Fakorede**                                     *fakorede@iastate.edu*
*Department of Computer Science*
*Iowa State University*

**Modeste Atsague**                                        *modeste@iastate.edu*
*Department of Computer Science*
*Iowa State University*

**Reviewed on OpenReview:** *https://openreview.net/forum?id=XgKO5fssnx*

## Abstract

Motivated by the vulnerability of feed-forward visual pathways to adversarial like inputs and the overall robustness of biological perception, commonly attributed to top-down feedback processes, we propose a new defense method AlignFix. We exploit the fact that natural and adversarially trained models rely on distinct feature sets for classification. Notably, naturally trained models, referred to as *weakM*, retain commendable accuracy against adversarial examples generated using adversarially trained models referred to as *strongM*, and vice-versa. Further these two models tend to agree more on their prediction if input is nudged towards correct class prediction. Leveraging this, AlignFix initially perturbs the input toward the class predicted by a naturally trained model, using a joint loss from both *weakM* and *strongM*. If this retains or leads to agreement, the prediction is accepted, otherwise the original *strongM* output is used. This mechanism is highly effective against leading SQA (Score-based Query Attacks) as well as decision-based and transfer-based black-box attacks. We demonstrate its effectiveness through comprehensive experiments across various datasets (CIFAR and ImageNet) and architectures (ResNet and ViT).

## 1 Introduction

Since the advent of adversarial attacks Szegedy et al. (2014), the field has seen an arms race between adversarial defenses and attacks. Defenses based on adversarial training Madry et al. (2018) have withstood the test of time. However, robust accuracy still needs improvement for reliable deployment. In realistic scenarios, attackers lack complete access to models, making black-box defense a practical choice and must be prioritized. However often these defenses only defend against Score based Query attacks (SQA) (like AAAChen et al. (2022), RND Qin et al. (2021)), while leaving transfer and decision-based attack surface open. Our work focuses on these practical challenges, proposing a defense that works well against all these black-box attacks.

We argue that simple feed-forward networks struggle with adversarial robustness due to the absence of an corrective feedback mechanism, a key component of biological perception Hawkins & Sandra (2004). Notably, Elsayed et al. (2018) found that under rapid, time-limited conditions ($\approx 74$ ms), where humans likely cannot engage these mechanisms, adversarial images also mislead human perception. Motivated by this, we propose incorporating a mechanism inspired by top-down feedback in biological perception to enhance the robustness of trained models.

|        | Nat'  | SAT   | TRADES | MART  |
|--------|-------|-------|--------|-------|
| Nat'   | 00.00 | **71.27** | **72.50** | **75.97** |
| SAT    | **82.74** | 51.61 | 62.22  | 64.19 |
| TRADES | **82.67** | 62.53 | 52.94  | 66.23 |
| MART   | **78.42** | 59.01 | 61.03  | 54.87 |

Table 1: Transfer accuracy of adversaries generated by different models. (ResNet-18, CIFAR-10, PGD-100 attack). Columns show models used for crafting the attack.

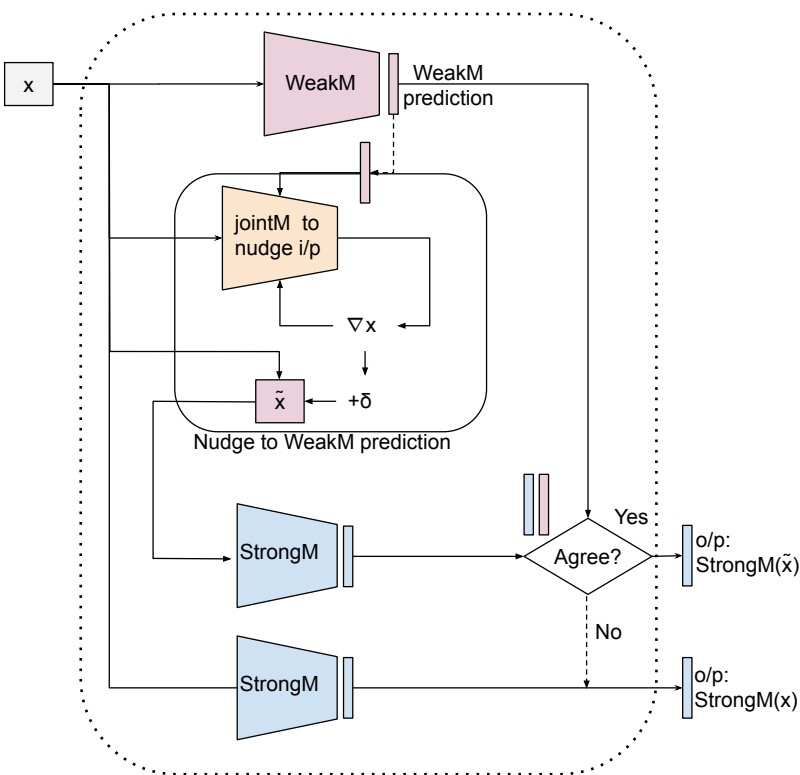

Figure 1: AlignFix Architecture: *WeakM* refers to a naturally trained model, *StrongM* refers to an adversarially trained model, and *jointM* refers to when the models are in parallel. Note that the input $x$ may be either clean or adversarially perturbed: e.g., via a transfer attack or an iterative black box attack using this (AlignFix) setup. AlignFix then computes $\tilde{x}$ by nudging $x$ toward the *WeakM* prediction using *jointM*.

A key challenge is to define a suitable feedback signal. Prior work Zhang & Zhu (2019) showed that adversarially trained models rely on shape, while naturally trained models rely on texture, indicating they attend to distinct and often orthogonal features. This difference makes it difficult for adversarial examples to transfer between them. Table 1 shows that adversarial examples crafted on a robust model (*strongM*) are more likely to be correctly classified by a naturally trained model (*weakM*), and the reverse also holds. Based on this, we use the disagreement between these models as an implicit corrective signal. We adjust the input by nudging it towards the naturally trained model's prediction using a joint loss from both models. If

this keeps or restores agreement, we accept the updated prediction, else we fall back to the robust model's original output. Figure 1 illustrates the architecture, and the full algorithm is described in Section 3.3.

To our knowledge, this is the first work to leverage a naturally trained model to strengthen the robustness of an adversarially trained model. Our contributions are as follows:

- We propose AlignFix, a simple method which incorporates a corrective perturbation mechanism to impart robustness under realistic black-box threats.

- We experimentally demonstrate that AlignFix improves robustness against realistic black box attacks, including: score-based, decision-based, transfer, and adaptive attacks.

Please note, while AlignFix is inspired by the top-down feedback and error correction principles observed in biological perception, it does not implement true feedback, and any direct comparison in mechanism would be inappropriate.

## 2 Background and Related Work

### 2.1 Preliminaries

We consider a $K$-class classifier $f$ parameterized by $\boldsymbol{\theta}$, which maps an input $x_i \in \mathcal{X}$ to its class label $y_i$. The model outputs logits $f_c(x_i, \boldsymbol{\theta})$ for each class $c$, and the predicted label is $y_{pred_i} = \arg\max_c f_c(x_i, \boldsymbol{\theta})$

We refer to naturally trained models as $weakM$ and adversarially trained models as $strongM$. Their predictions are denoted $y_{weakM_i}$ and $y_{strongM_i}$ respectively. During defense, we perturb the input using both models jointly by summing their cross-entropy losses; we denote this configuration as $jointM$.

In adversarial settings, the input $x_i$ is perturbed to $x_i'$ to induce misclassification. Perturbations are constrained to $B_\epsilon[x_i] = x_i' : \|x_i' - x_i\|_p < \epsilon$, where $\|.\|_p$ is the $\ell_p$ norm and $\epsilon$ is the perturbation budget. The projection operator $\prod$ clips $x'$ to keep it within $B_\epsilon[x_i]$.

We focus on the $\ell_\infty$-norm as it is the standard in adversarial robustness research, offering a well-established benchmark with imperceptible and efficiently computable perturbations. Following convention, we set $\epsilon = 8/255$ for CIFAR-10 and $\epsilon = 4/255$ for ImageNet.

### 2.2 Adversarial attacks and defenses

This subsection summarizes adversarial attacks and defenses most relevant to our work. For a broader survey, see Akhtar et al. (2021).

**Adversarial attacks:** Common white-box attacks include FGSM Goodfellow et al. (2014), PGD and targeted-PGD Kurakin et al. (2018); Madry et al. (2018), and their variants such as AutoPGD (APGD) Croce & Hein (2020b), FAB Croce & Hein (2020a), and C&W Carlini & Wagner (2017). These attacks use the gradient of the loss with respect to the input $x_i$ to iteratively estimate a perturbation direction that maximizes loss in a local neighborhood:

$$x_i'^{t+1} \leftarrow \prod_{B_\epsilon[x_i]} (x_i'^t + \alpha \cdot \text{sgn}(\nabla_{x_i'^t} l(x_i'^t, y_i))) \tag{1}$$

Variants differ in their choice of loss function, step size, and update rules. These are considered the strongest attacks, as the adversary has full access to the model parameters.

In real-world scenarios, attackers typically lack full model access. If they can query class confidence scores, they use Score-based Query Attacks (SQA) to iteratively craft adversaries. The Square attack Andriushchenko et al. (2020) is a leading example: it perturbs random square regions by $\pm 2\epsilon$ and retains changes that reduce confidence in the true class. Other SQA methods include Bandit Ilyas et al. (2018b),

SimBA Guo et al. (2019), ZOO Chen et al. (2017), SignHunter Al-Dujaili & O'Reilly (2019), and NES Ilyas et al. (2018a).

In decision-based (hard-label) attacks, only the predicted class is available. Notable examples include SPSA Uesato et al. (2018), HopSkipJump Chen et al. (2020), RayS Chen & Gu (2020), and others Ma et al. (2021); Shukla et al. (2021); Cheng et al. (2018); Brendel et al. (2018).

In transfer attacks, adversaries craft inputs using white-box attacks on a surrogate model. Due to the transferability of adversarial examples Szegedy et al. (2014), such attacks often succeed when the surrogate shares architecture or training data. When combined with decision-based attacks, they greatly reduce query counts Sitawarin et al. (2024).

**Adversarial defense:** Adversarial Training (AT), which trains models on adversarial examples, remains the most effective defense. Key methods include SAT Madry et al. (2018) and TRADES Zhang et al. (2019), with several extensions such as MART Wang et al. (2019), GAIRAT Zhang et al. (2020), HE Pang et al. (2020); Fakorede et al. (2023), MAIL Liu et al. (2021), and AWP Wu et al. (2020); Yu et al. (2022).

Most black-box defenses neglect decision-based and transfer attacks, leading to a false sense of robustness. Such defenses can often be bypassed by training a surrogate model. The need to guard against transfer attacks is highlighted by Szegedy et al. (2014); Sitawarin et al. (2024).

## 2.3 Related work

AlignFix (ours) is an adaptive defense, modifying the input at inference to correct potential adversarial perturbations. While several methods also perform input nudging, their goals differ. Wu et al. (2021) maximize the total cross-entropy, which Croce et al. (2022) show can hurt accuracy near decision boundaries. Shi et al. (2021) use self-supervised signals, whereas AlignFix leverages the disagreement between $strongM$ (adversarially trained) and $weakM$ (naturally trained), incorporating guidance from $weakM$. Tao et al. (2022) and Li et al. (2023) apply nudging during training and are not adaptive defenses.

RND Qin et al. (2021) adds Gaussian noise to resist SQA attacks. AAA Chen et al. (2022), closest to ours, adaptively alters logits to mislead SQA attacks but remains vulnerable to transfer attacks. AlignFix, by contrast, increases the effective decision boundary of $strongM$, providing robustness beyond just SQA. We treat AAA as our key baseline.

## 3 Methodology

### 3.1 Motivation from Biological Perception

Despite extensive research, neural networks still lack the robustness of biological perception. Adversarially trained models extract more human aligned features Zhang & Zhu (2019), but they remain vulnerable to real world transformations such as rotation and translation Engstrom et al. (2019). This raises a natural question:

***Can feed-forward neural networks be adversarially robust, or are they intrinsically vulnerable?***

Elsayed et al. (2018) showed that adversarial images can fool humans when presented briefly (71ms), but not when given more time (2s). This suggests that the human brain corrects perception using top-down and lateral feedback. Guo et al. (2022) compared neural activity in primates with representations in ResNet-50. Surprisingly, adversarially trained ResNet-50 was more stable to attacks than the primate visual system under black-box perturbations. Yet, humans (and likely primates) are not easily fooled, implying some form of error correction is present in the brain.

### 3.2 Key Insight from Model Disagreement

While making a prediction, if an oracle could identify the correct class, we could nudge the input toward the correct class to undo adversarial perturbations. We propose using the disagreement between $strongM$ (adversarially trained) and $weakM$ (naturally trained) models as a proxy for such an oracle.

---

**Algorithm 1** AlignFix

> **Inputs:** $x_i$, *strongM*, *weakM*, *s_size*
> **Output:** Prediction logits for the input $x_i$

---

1: $logitW \leftarrow weakM(x_i)$
2: Obtain $\tilde{x}_i$ by nudging $x_i$ toward $\arg\max(logitW)$ using Equation (2) with step size *s_size* for 1 step
3: $\tilde{logitS} \leftarrow strongM(\tilde{x}_i)$
4: **if** $\arg\max(\tilde{logitS}) = \arg\max(logitW)$ **then**
5:      **return** $\tilde{logitS}$
6: **end if**
7: **return** $strongM(x_i)$

---

As shown in Table 1, adversarial examples transfer poorly between these models. Attacks are most successful in the white-box setting. When attacked using adversaries generated from naturally trained models, adversarially trained models show high accuracy. Conversely, naturally trained models maintain the highest accuracy even when attacked using adversarially trained models. This is likely because the two models rely on distinct features—texture for naturally trained models and shape for adversarially trained ones.

Zhang & Zhu (2019) further showed that naturally trained models can still classify shuffled image tiles, while adversarially trained models fail. This was quantified in their Figure 7 (borrowed in Figure 6 in Appendix C), where AT-CNN accuracy drops sharply under patch-shuffling (which destroys shape but preserves local textures), whereas standard CNNs remain largely unaffected. Their saliency map analysis (Figure 2, borrowed in Figure 5) further shows that AT-CNNs focus on object contours, while standard CNNs attend to dispersed texture regions. In absence of adversarial training, the model does not have incentive to learn robust features Tsipras et al. (2018). These findings reinforce that adversarially trained models are shape-biased and robust to texture perturbations. We find that combining the two models yields stronger robustness.

We assume that when the models agree, the prediction is likely correct. Since both models are trained for accuracy, simultaneous errors are rare. We experimentally validated this assumption for CIFAR-10 using the ResNet-18 model. We found that when an adversary example is both crafted and nudged using *jointM*, then nudging towards the correct class increases the agreement between the two models' predictions from 34.98% to 49.20%, while if they are nudged towards a random class, the agreement decreases to 16.99%.

### 3.3 AlignFix

We present AlignFix in Algorithm 1. First, we compute the logits $logitW$ returned by *weakM* on the input $x_i$ (Step 1). Then, we nudge the input one step toward the predicted class $\arg\max(logitW)$ using Equation (2) (Step 2). This step uses a joint loss from both *weakM* and *strongM*:

$$\tilde{x}_i \leftarrow \prod_{B_\epsilon[x_i]} \left( x_i - s\_size \cdot \text{sign}(\nabla_{x_i^t} l(x_i^t, y_i^o)) \right) \tag{2}$$

$$\text{where} \quad l = l_{ce}(strongM(x_i), y_i^o) + l_{ce}(weakM(x_i), y_i^o) \tag{3}$$

$$y_i^o = \arg\max_k (logitW_k) \tag{4}$$

Here, $\prod$ denotes the projection operator that ensures $\tilde{x}_i$ stays within the allowed perturbation budget $\epsilon$. Next, we compute $\tilde{logitS}$ from *strongM* using the nudged input $\tilde{x}_i$ (Step 3). If the prediction $\arg\max(\tilde{logitS})$ matches $\arg\max(logitW)$, we return $\tilde{logitS}$ as output (Step 5). Otherwise, we discard the nudging and return the prediction from *strongM* on the original input $x_i$ (Step 7).

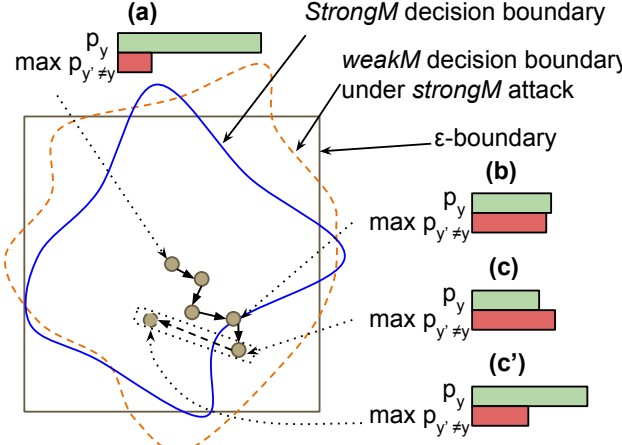

Figure 2: Illustration of the adaptive defense mechanism in AlignFix. Here, $y$ denotes the correct class and $y'$ denotes the incorrect class with the highest logit value. The logit for the correct class, $p_y$, is shown in green, while the maximum logit among incorrect classes, which may vary across steps, is represented by $max p_{y' \neq y}$ and is shown in red. Please note both the logits are logits from AlignFix, which always use *strongM* to output final logit. See text for further explanation.

### 3.3.1 Robustness through AlignFix

Figure 2 illustrates how AlignFix defends against adversarial perturbations by leveraging model disagreement. A clean input $x_i$ at point (a) is correctly classified by both models. An attacker perturbs it toward misclassification, reaching point (b), but as long as both models agree, AlignFix nudges the input toward the agreed class and returns the prediction. When the models begin to disagree, two cases arise: *weakM* will make the correct prediction (case A) or not (case B). In case A, shown at point (c), *weakM* remains correct while *strongM* is fooled. AlignFix nudges the input in the direction favored by *weakM*, using both models jointly. This leads to point (c'), where if the models now agree, AlignFix returns the corrected prediction from *strongM*. This effectively expands the decision boundary of the robust model, allowing recovery from misclassification. If not, it falls back to the original output of *strongM*. For case B, when *weakM* makes an incorrect prediction, we rely on robustness of *strongM*. Since it is hard to fool *strongM*, the nudging done in the incorrect direction would, often, not be enough for *strongM* to cause misclassification, especially under realistic black-box scenario.

This mechanism increases the effective decision boundary of *strongM*, making black-box attacks less effective. Figure 3 shows this effect on the Square attack. While the attack fools *strongM* and *weakM* individually, AlignFix prevents failure by letting one model correct the other's error. This frustrates the attack loop and maintains accuracy. Similar robustness is observed against SPSA and RayS. AlignFix also performs well under transfer attacks due to the inclusion of *strongM*, which learns robust features.

## 3.4 Design Choices

Since *strongM* and *weakM* rely on different features, one option was to nudge the input only when their predictions disagree. This would expose only *strongM* to the attacker until needed. However, unless the input is corrected continuously, this approach reduces AlignFix's effectiveness on SQA attacks by nearly 10 percent. Therefore, we always apply nudging and discard it only when it fails to restore agreement.

Logits are always returned from *strongM*, keeping *weakM*'s confidence hidden. Because *weakM* is easier to fool, relying on *strongM* when they disagree improves robustness. We found that using *weakM* for output drops Square attack accuracy from 83.5 percent to 71.8 percent on CIFAR-10.

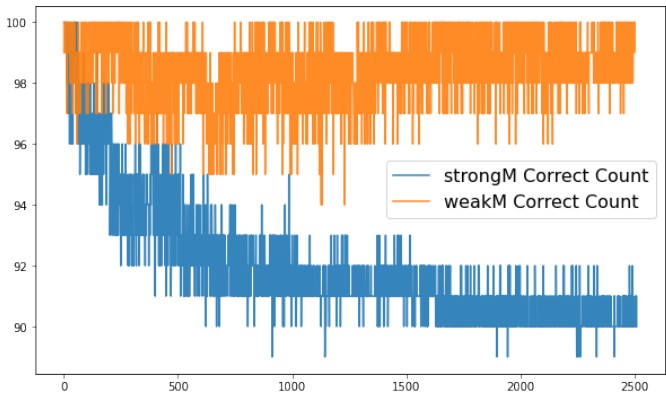

Figure 3: AlignFix mitigates the Square attack by combining *strongM* and *weakM*. While the attack individually fools both models over 2500 iterations, their combination in AlignFix, prevents simultaneous error preventing failure

.

To nudge the input, two parameters are required: (a) *n_steps*, the number nudging steps, and (b) *s_size*, the size of each step. The product $n\_steps \times s\_size$ determines the total nudging magnitude. We found that multiple small steps could be approximated by a single large step, so we fix $n\_steps = 1$. Since *strongM* is trained with a perturbation budget of $\epsilon = 0.031$ and *weakM* with $\epsilon = 0$, the appropriate nudging lies roughly in between, around $\epsilon/2$. Based on experiments on CIFAR-10 with ResNet-18 (Appendix Section B, Ablation Study), we fix $s\_size = 0.02$. As shown in Table 9 in Appendix, this value trades off transferability and black-box robustness. For ImageNet, where $\epsilon = 4/255$, we use $s\_size = 0.01$, which yields consistent performance.

## 4 Experiments

### 4.1 Setup

We evaluated AlignFix on CIFAR-10 and ImageNet datasets [1]. For fine-tuning and ablation, we used ResNet-18. Following our baseline AAA-linear Chen et al. (2022), we reported results using WideResNet-28-10 for CIFAR-10 and WideResNet-50 for ImageNet. Appendix D provides model sources and training details.

For Square attacks, we used the official implementation of AutoAttack (`https://github.com/fra31/auto-attack`) and set $p_{init} = 0.05$, consistent with AAA-linear. SPSA was run for 100 iterations with perturbation size 0.001, learning rate 0.01, and 128 samples per gradient estimate. For RayS, we used the official code (`https://github.com/uclaml/RayS`) with 1K and 10K queries.

### 4.2 Results

In this section, we present results for black-box attacks on CIFAR-10 and ImageNet.

#### 4.2.1 SQA attacks

Table 2 shows the results for black-box SQA attacks on WideResNet-28-10 models. Attack parameters are given in Appendix E. We evaluated on the first 1K samples from CIFAR-10. RND results are taken from Chen et al. (2022).

AlignFix-SAT, using the SAT model Madry et al. (2018) as *strongM*, outperforms all baselines including AAA-Linear on the Square attack. It also performs better on NES, while SignHunter and Bandit slightly favor AAA-Linear. SimBA results are similar for both, despite higher natural accuracy of AAA-Linear.

---

[1]Code is available in supplement and at: `https://github.com/aknirala/AlignFix`

| Defense | Nat' | Accuracy on SQA Attack (# queries = 100/2500) | | | | |
|---|---|---|---|---|---|---|
| Methodology | Acc' | Square | SignHunter | SimBA | NES | Bandit |
| Undefended | 94.78 | 39.7/00.2 | 42.3/00.0 | 73.5/35.6 | 68.8/05.0 | 49.9/01.3 |
| RND | 91.05 | 60.8/49.1 | 61.0/47.8 | 76.4/64.3 | 86.2/68.2 | 70.4/41.6 |
| AAA-Linear | **94.84** | 83.4/79.8 | 84.2/83.0 | 86.4/84.5 | 84.6/71.0 | 86.7/82.8 |
| SAT | 85.83 | 76.9/60.5 | 74.9/56.6 | 84.1/80.4 | 83.3/75.4 | 78.7/66.2 |
| AlignFix-SAT | 90.30 | 85.7/84.3 | 80.5/79.4 | 86.0/83.6 | 87.3/73.2 | 85.0/81.9 |
| TRADES | 86.40 | 77.1/61.2 | 74.9/57.0 | 86.2/82.6 | 85.4/74.8 | 80.3/66.2 |
| AlignFix-TRADES | 91.65 | 87.4/85.8 | 81.0/79.0 | 86.6/85.8 | 87.9/74.7 | 85.7/83.0 |
| AWP | 85.36 | 75.9/62.7 | 74.0/60.0 | 84.1/80.4 | 83.4/75.2 | 79.1/68.6 |
| AlignFix-AWP | 90.00 | 86.9/85.0 | 79.7/77.7 | 86.4/84.9 | 87.0/75.1 | 85.1/82.5 |
| AWP_E | 88.25 | 81.3/67.8 | 79.5/63.4 | 87.2/84.4 | 86.9/79.9 | 83.4/72.5 |
| AlignFix-AWP_E | 91.80 | 87.8/87.5 | 83.8/82.5 | 88.0/85.8 | 88.4/77.5 | 86.0/83.7 |
| WANG23 | 92.44 | 86.5/75.5 | 85.0/71.6 | **92.1/89.1** | 91.5/84.8 | 87.8/79.7 |
| AlignFix-WANG23 | 94.40 | **91.4/90.9** | **87.2/85.6** | 91.0/89.5 | **92.1/81.6** | **89.8/88.0** |

Table 2: AlignFix performance compared to baselines on SQA attacks for the CIFAR-10 dataset with a perturbation budget of $\ell_\infty = \frac{8}{255}$ (queries = 100/2500). WideResNet-28-10 is used for all models.

To test generality, we used other robust models as *strongM*: TRADES Zhang et al. (2019), AWP, AWP_E Wu et al. (2020), and WANG23 Wang et al. (2023). With these, AlignFix consistently achieves the highest accuracy across all attacks. For SimBA, we used the SimBA-DCT variant from `https://github.com/cg563/simple-blackbox-attack`, scaling perturbations to the allowed maximum. This variant avoids the degenerate case where one-pixel changes fail to fool robust models.

| Attack | Undefended | SAT | RND | AAA-Linear | AlignFix (Ours) |
|---|---|---|---|---|---|
| ACC(%) | **78.48** | 68.46 | 75.32 | **78.48** | 72.35 |
| Square | 55.40/10.90 | 61.90/54.40 | 58.67/50.54 | 64.35/63.96 | **67.05/64.95** |
| SignHunter | 62.25/17.30 | 62.65/58.25 | 59.36/52.98 | **71.75 /71.25** | 67.25/64.80 |
| SimBA | 70.65/57.35 | 66.40/64.80 | 66.36/63.27 | 70.80/66.20 | **72.75/69.90** |
| NES | 76.15/59.35 | 67.15/64.65 | 71.33/66.05 | **76.60/70.25** | 70.80/66.25 |
| Bandit | 62.60/27.65 | 64.70/59.45 | 65.15/61.38 | **69.70/69.20** | 69.10/67.95 |

Table 3: AlignFix performance compared to baselines on SQA attacks for ImageNet dataset, with a perturbation budget of: $\ell_\infty = \frac{4}{255}$ (#query = 100/2500). WideResNet-50 is used for all models.

Table 3 presents ImageNet results. AAA-Linear evaluated Square attack on 1K randomly selected samples (one per class), correctly classified by the naturally trained model. They then scaled the accuracy by 78.48%. In this work, we evaluate and report results (including re-evaluation of our baseline AAA) on the first 2K samples from the validation set, selected by sorting image filenames alphabetically. This yields consistent yet random-like sampling, as class labels are randomly distributed with respect to filename order. While this unfiltered setting slightly favors AAA, AlignFix outperforms AAA-Linear for Square and SimBA attacks.

**Effect of $p_{init}$ in Square Attack:** AAA-Linear used $p_i nit = 0.05$ for CIFAR-10 and 0.3 for ImageNet, matching the original setup for naturally trained models. Later works use 0.8 to better attack robust models. We found that this change severely reduces AAA-Linear accuracy, while AlignFix remains stable (Figure 4).

| SurrogM | Natural | SAT | AAA | AlignFix |
|---|---|---|---|---|
| (ResNet-18) | (WideResNet-28-10) | | | |
| AT | 73.17 | 64.85 | 73.17 | 69.33 |
| Natural | 16.91 | 85.03 | **16.91** | 79.47 |
| *jointM* | 15.36 | 78.92 | **15.36** | 72.95 |

Table 4: Transfer attack accuracies on CIFAR-10, using ResNet-18 architecture as surrogate Model. We note that, AAA-Linear accuracy drops considerably.

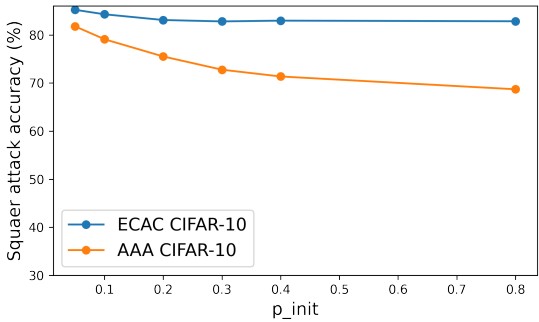

Figure 4: Square attack accuracy of AAA-Linear for different values of *p_init* (fraction of pixels changed every iteration.) for CIFAR-10

| Surrogate M (WRN-50) | Natural | SAT | AAA | AlignFix |
|---|---|---|---|---|
| | | (WideResNet-50) | | |
| AT | 67.05 | 40.65 | 67.05 | 48.95 |
| Natural | 00.01 | 67.15 | **00.01** | 61.90 |
| *jointM* | 00.00 | 59.10 | **00.00** | 55.70 |

Table 5: Transfer accuracy of adversaries generated for ImageNet. We used the same component models that are used in the defense.

### 4.2.2 Transfer and Decision-based Attacks

**Transfer attacks** Most black-box defenses overlook the threat posed by transfer attacks, even though a determined adversary can easily train a surrogate model and launch effective attacks without direct access to the target. For CIFAR-10, we used a ResNet-18 model to generate PGD-20 adversaries and tested them on WideResNet-28-10 models, i.e., the surrogate is a much smaller network.

As shown in Table 4, AlignFix achieves the highest worst-case robustness, especially when adversaries are crafted using naturally trained surrogates where other defenses fail. For ImageNet, despite using the same models (due to high computation cost) as AlignFix for crafting the attack, AlignFix retains significantly higher robustness compared to AAA (Table 5).

| Models | RayS (1K/10K queries) | SPSA |
|---|---|---|
| Undefended | 22.30/00.10 | 00.00 |
| AT | 71.40/59.90 | 62.40 |
| AAA-Linear | 58.50/55.10 | 70.10 |
| AlignFix (ours) | **72.00/66.60** | **79.00** |

Table 6: AlignFix performance compared to baselines, using WideResNet-28-10, on decision-based attacks for CIFAR-10, with the $\ell_\infty$ perturbation of: $\frac{8}{255}$.

**Decision-based attacks** We evaluated AlignFix on SPSA and RayS attacks for CIFAR-10 (Table 6). While both AlignFix and AAA show good robustness against SPSA, RayS proves more effective against AAA due to its strategy of starting from a high-perturbation misclassified point. In contrast, AlignFix's dynamic decision boundary offers consistently higher robustness in both cases.

### 4.2.3 Adaptive Attacks

AlignFix is an adaptive defense, and as noted by Croce et al. (2022), it is imperative to evaluate it using adaptive attacks. The principle of designing adaptive attacks is to exploit the inner weakness of a given defense. Since our method relies on the two component models, which use different sets of features, attacks that attempt to remove both features are likely to bring its accuracy down. Under our setting, we consider

the case where the attacker knows the overall AlignFix architecture (Figure 1) but lacks access to the internal *strongM* and *weakM* models weights. In a realistic black-box setting, the attacker can query the deployed AlignFix model and can also train surrogate models to approximate its behavior. For CIFAR-10, we used ResNet-18 models trained with standard and adversarial training as surrogates.

---

**Algorithm 2** Adaptive Attack for AlignFix Defense

---

**Inputs:**

$(x, y)$: Input and label pair

$surrStrongM$, $surrogWeakM$: Surrogate models

$AlignFix$: Deployed AlignFix model

$s\_size$, $\epsilon$: Parameters for AlignFix defense and perturbation budget

$pgd\_itrs$, $pgd\_s\_size$: PGD attack iterations and step size

$pgdAtk(input, label, model, pert\_bdgt, pgd\_itrs, pgd\_s\_size, do\_t)$: PGD attack function; $do\_t$ indicates if the attack is targeted

**Output:** $x'$ such that $AlignFix(x') \neq y$ or $FAILURE$ if not found

---

1: $x_t \leftarrow pgdAtk(x, y, surrStrongM, \epsilon + s\_size, pgd\_itrs, pgd\_s\_size, \text{False})$
2: $y_t \leftarrow \arg\max(surrStrongM(x_t))$
3: $do\_t \leftarrow \text{True}$
4: **if** $y_t == y$ **then**
5:     $do\_t \leftarrow \text{False}$
6: **end if**
7: $x' \leftarrow x$
8: $jointM \leftarrow$ Combine $surrStrongM$ and $surrogWeakM$
9: **for** $itr = 1$ to $pgd\_itrs$ **do**
10:     $x' \leftarrow pgdAtk(x', y_t, jointM, \epsilon, 1, pgd\_s\_size, do\_t)$
11:     **if** $\arg\max(AlignFix(x')) \neq y$ **then**
12:         **return** $x'$
13:     **end if**
14: **end for**
15: **return** $FAILURE$

---

The goal of an adaptive attack is to exploit structural weaknesses in the defense. For AlignFix, two such cases arise: (a) both *strongM* and *weakM* agree on an incorrect prediction after nudging, or (b) the two models disagree, and *strongM* alone makes an error. Case (a) is particularly damaging, as AlignFix returns the incorrect shared prediction. In case (b), AlignFix discards nudging and falls back on *strongM*, which, if wrong, also fails the defense. To target both cases, we devised a two-stage PGD-based strategy (Algorithm 2):

- **Step 1: Find a vulnerable class using *strongM* (Lines 1–3)**: We first run an untargeted PGD attack on the surrogate strong model *surrStrongM* with an expanded budget of $\epsilon + s\_size$ (Line 1). This yields an adversarial example $x_t$. We extract the predicted label $y_t$ from *surrStrongM* on $x_t$ (Line 2). If $y_t$ still matches the true label $y$ (Line 4), we skip targeting and fall back to untargeted PGD later by setting $do\_t$ to $False$ (Lines 5–6).

- **Step 2: Launch a targeted PGD attack using *jointM* (Lines 7–14)**: We initialize $x' \leftarrow x$ (Line 7) and define *jointM* by combining the surrogate strong and weak models (Line 8). Then, for each PGD iteration (Line 9), we update $x'$ by running a targeted (or untargeted, depending on $do\_t$) PGD attack on *jointM*, aiming for class $y_t$ (Line 10).

- **Step 3: Query the actual AlignFix model (Lines 11–13)**: After each update, we check whether AlignFix misclassifies $x'$ (Line 11). If it does, we return $x'$ as a successful adversarial example (Line 12). If not, the loop continues.

- **Step 4: Fallback (Line 15)**: If the attack fails after all iterations, we return $FAILURE$ (Line 15). This probes both failure modes of AlignFix: (a) when both models agree incorrectly, and (b) when fallback to *strongM* fails.

| Surrogate Model Accuracy | | | **AlignFix** |
|---|---|---|---|
| *strongM* | *weakM* | AlignFix | **Accuracy** |
| 50.59 | 00.00 | 40.00 | **60.91** |

Table 7: AlignFix Accuracy on Adaptive Attack.

To construct an effective adaptive attack, we utilized surrogate models trained on analogous datasets. For CIFAR-10, we trained ResNet-18 using standard and Madry's Madry et al. (2018) method as our surrogate models. The initial budget is $\epsilon + s\_size$, where $s\_size$ is the nudging used by AlignFix as defense (0.02 for CIFAR-10). As shown in Table 7, AlignFix maintains high accuracy even under such informed attacks. Notably, even the surrogate AlignFix model, under white-box attack, retains 40% accuracy—highlighting the robustness imparted by *strongM*.

### 4.2.4 Results on ViT architecture

| Models (ViT) | Nat' | Square (100/2.5k) | RayS (1k/10k) | SPSA |
|---|---|---|---|---|
| Natural | **91.8** | 42.5/00.1 | 18.7/00.3 | 06.6 |
| SAT | 76.4 | 65.3/52.7 | 59.5/51.2 | 64.8 |
| AlignFix-SAT | 80.1 | **75.8/74.4** | **61.8/58.6** | **76.7** |
| TRADES | 80.6 | 69.7/56.4 | 63.7/53.8 | 69.6 |
| AlignFix-TRADES | 84.8 | **79.9/77.9** | **65.7/61.5** | **80.3** |

Table 8: AlignFix CIFAR-10 accuracy with ViT architecture.

To evaluate AlignFix's generality across architectures, we trained Vision Transformer (ViT) models on CIFAR-10 using the code from Mo et al. (2022). As shown in Table 8, AlignFix consistently improves the robustness of *strongM* for both SAT and TRADES, confirming its applicability beyond convolutional networks.

## 5 Limitations

AlignFix is designed for realistic scenarios of black-box settings, including decision-based and transfer attacks. The AlignFix defense, by always outputting the *strongM* logit, theoretically constrains the total perturbation to the sum of the adversarial attack's perturbation ($\epsilon$) and the defense's corrective nudge (*step_size*). This may provide a degree of white-box robustness, however comprehensive empirical evaluation against white-box attacks is outside the scope of this paper. Moreover, like other adversarial robustness methods (e.g., AAA, AT), this work focuses specifically on adversarial robustness under $\ell_\infty$ threats. Adversarial training on its own does not necessarily extend to all types of robustness, like robustness to other transformations such as rotation as shown by Engstrom et al. (2019). Handling them requires *strongM* and *weakM* to be trained for spatial robustness. This is an orthogonal challenge that can be incorporated into AlignFix by substituting spatially robust base models.

**Increased Inference Time:**

- AlignFix incurs 5× overhead due to one backward and two forward passes per input. A simple ResNet-18 model takes $\approx 1.6$ seconds to classify the entire CIFAR-10 test set, with a batch size of 500 using an RTX-2080 Ti graphics card, whereas AlignFix takes $\approx 8.3$ seconds. This is a tradeoff for strong robustness under black-box threats, including decision-based and transfer attacks where prior defenses fail.

- As discussed in Section 3.1, biological perception relies on corrective feedback Hawkins & Sandra (2004). Guo et al. (2022); Elsayed et al. (2018) attributed the role of feedback for robustness in biological perception. While adversarially robust feed-forward models are ideal, current evidence suggests this may be "fundamentally limited". Despite added cost, AlignFix remains practical e.g., classifying $10K$ CIFAR-10 samples took $8.3s$ on an RTX 2080 Ti. This real-time performance is

acceptable in safety-critical settings like medical triage, surveillance, parts of autonomous driving like street sign detection, or offline robust curation.

- Mitigation strategies include gradient approximations, model distillation, or hardware acceleration.

**Broader Impact Concerns:** We acknowledge that, while AlignFix is developed as a defense, insights into model vulnerabilities can inform stronger attacks, raising dual-use risks. Moreover, the increased computational cost due to multiple model evaluations may contribute to higher energy consumption, potentially limiting accessibility and raising environmental concerns. These tradeoffs underscore the importance of developing energy efficient computational approaches, such as spiking neural networks Roy et al. (2019), as well as broadly applicable and interpretable defenses.

## 6 Conclusion and Discussion

Despite progress in adversarial robustness, current models remain vulnerable to practical black-box attacks. Inspired by the stability of biological perception, we introduced AlignFix, a defense that combines naturally and adversarially trained models to perform correction via prediction disagreement. AlignFix improves robustness across diverse attack types without requiring access to model internals, making it suitable for realistic deployment scenarios.

This work highlights the value of incorporating corrective feedback principles into model design and points toward future systems that may integrate memory or structured world knowledge to further enhance robustness.

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

## A  Model agreement and correct prediction

We work with the assumption that when the two models agree, their agreed prediction tends to be correct. While this is intuitive, as the models are trained to make the correct prediction, we further analyze how and when this assumption holds.

Our assumption is based on the intuition that the chance of *StrongM* and *WeakM* predicting the same wrong class is low. Formally, let the accuracy of *StrongM* be $p_s$ and *WeakM* $p_w$, their predictions be $S_p$ and $W_p$, and $y$ be the correct prediction. Assume their predictions are independent and equally likely to predict any of the wrong labels, then

$$P(S_p = y')_{y' \neq y} = \frac{(1 - p_s)}{(C - 1)}, \tag{5}$$

$$P(W_p = y')_{y' \neq y} = \frac{(1 - p_w)}{(C - 1)} \tag{6}$$

where $C$ is the number of classes. We have:

$$\frac{P(correct|agree)}{P(incorrect|agree)} = \frac{P(correct, agree)}{P(incorrect, agree)} \tag{7}$$

$$= \frac{P(S_p = y, W_p = y)}{\sum_{y' \neq y} P(S_p = y', W_p = y')} \tag{8}$$

$$= (C - 1) \frac{p_s}{(1 - p_s)} \frac{p_w}{(1 - p_w)}. \tag{9}$$

Therefore, the assumption holds when $p_s$, $p_w > 50\%$ for binary classification and could hold even when $p_s$, $p_w$ is small for multi-class classification.

Further, as noted in the main paper, we experimentally verified this. For CIFAR-10 using the ResNet-18 model, we found that when an adversary is both crafted and nudged using both *strongM* and *weakM* in parallel (i.e., *jointM*), then nudging towards the correct class increases the agreement between the two models' predictions from 34.98% to 49.20%, while if they are nudged towards a random class, the agreement decreases to 16.99%.

## B  Ablation Study

We conducted ablations for CIFAR-10 by varying the nudging step size $s\_size$, the number of nudging steps $n\_step$, the weighting between *strongM* and *weakM* in *jointM*, and by testing different *strongM* models.

| | Value for $s\_size$ | | | | |
|---|---|---|---|---|---|
| Accuracy | 0.015 | 0.018 | 0.020 | 0.022 | 0.025 |
| Natural | 88.60 | 88.70 | 88.70 | 89.10 | **89.40** |
| Square (1k iterations) | 81.30 | 81.60 | 81.40 | **82.80** | 82.00 |
| RayS (1k iterations) | 72.00 | **72.30** | 72.00 | 72.00 | 71.30 |
| Transfer (using *jointM*) | **69.60** | 68.00 | 67.30 | 66.70 | 65.30 |

Table 9: Effect on AlignFix accuracy for different values of $s\_size$, under various attacks. The results are for the first 1000 samples of CIFAR-10.

Increasing $s\_size$ improves robustness to SQA attacks but reduces performance on transfer attacks, as seen in Table 9. For $n\_step$, using more steps slightly boosts natural accuracy but reduces SQA accuracy and increases runtime (Table 10), making one-step nudging a practical choice. To construct *jointM*, we experimented with different weights on *strongM* and *weakM* losses. Table 11 shows that setting $\alpha = 0.5$ yields a good balance across natural accuracy, SQA, and transfer attacks. Finally, while adaptive nudging methods like AutoPGD could improve performance by tailoring perturbations per sample, their high computational cost makes our simpler one-step nudging a more practical choice.

| $n\_steps$ | Size of each step | Natural Accuracy | Square Attack Accuracy | Square Attack Time (sec) |
|---|---|---|---|---|
| 1 | 0.020 | 87.8 | 82.4 | 509.6 |
| 2 | 0.011 | 89.3 | 79.6 | 814.4 |
| 3 | 0.007 | 89.3 | 78.6 | 1123.1 |

Table 10: Effect on AlignFix accuracy for different values of $n\_steps$, where total perturbation by nudging was clipped at 0.02. The results are for the first 1000 samples of CIFAR-10.

| $alpha$ | Natural Accuracy | Square Attack Accuracy | Transfer Attack Acc' using surrog' | | |
|---|---|---|---|---|---|
| | | | $strongM$ | $weakM$ | $jointM$ |
| 0.0 | 91.8 | 80.3 | 70.84 | 77.88 | 70.76 |
| 0.1 | 91.4 | 82.7 | 69.88 | 78.94 | 71.49 |
| 0.2 | 91.0 | 84.0 | 69.48 | 78.94 | 71.80 |
| 0.3 | 91.1 | 84.3 | 69.51 | 79.31 | 71.75 |
| 0.4 | 90.9 | 84.8 | 69.38 | 79.54 | 71.79 |
| 0.5 | 90.9 | 84.5 | 69.31 | 79.45 | 72.20 |
| 0.6 | 90.4 | 85.1 | 68.78 | 79.49 | 72.12 |
| 0.7 | 90.3 | 85.5 | 68.72 | 79.66 | 72.41 |
| 0.8 | 90.3 | 85.2 | 68.56 | 79.77 | 72.50 |
| 0.9 | 89.8 | 84.4 | 68.78 | 80.11 | 72.44 |
| 1.0 | 86.6 | 81.7 | 65.62 | 84.44 | 77.11 |

Table 11: Effect on AlignFix accuracy on CIFAR-10 for different values of $\alpha$ as used to define $jointM$. AlignFix uses WideResNet-28-10 models, while surrogate models use ResNet-18 architecture.

## C  AT-CNN are shape biased

We have leveraged the observation that adversarially trained models, which we refer to as *strongM*, are shape biased, while naturally trained models, referred to as *weakM*, are texture biased. This observation is grounded in the findings of Zhang & Zhu (2019), who conducted systematic qualitative and quantitative comparisons between adversarially trained CNNs (AT-CNNs) and normally trained CNNs. Below is some of their key observation:

- Qualitatively, Figure 2 in their paper (borrowed in Figure 5), based on SmoothGrad salience maps, shows that AT-CNNs consistently highlight object contours across clean, saturated, and stylized images, whereas standard CNNs produce noisy, texture-focused maps.

- Quantitatively:
  - Their, Table 2 shows that on stylized images (which distort textures but preserve shape), AT-CNNs outperform standard CNNs by a wide margin, confirming texture invariance.
  - Their Figure 7 (borrowed in Figure 6) demonstrates a steep drop in AT-CNN accuracy under patch-shuffling (which preserves local textures but destroys shape), while standard CNNs remain largely unaffected, confirming AT-CNNs' reliance on shape.

## D  Model source and training details used for defense

We provide the source of the WideResNet models in Table 12. For ResNet-18 and for cases where the corresponding model is not present on RobustBench (i.e., *strongM* for CIFAR-10), we trained the model locally. We used Madry's et al. Madry et al. (2018) method to train all the adversarially robust models for CIFAR-10, which are used as *strongM*. In line with the settings used in the literature Wang et al. (2019); Liu et al. (2021), all the base models (i.e., those included in Table 4 as well) have been trained for 120 epochs using mini-batch gradient descent with an initial learning rate of 0.01 (0.1 for WideResNet), which was decayed by a factor of 10 at epoch 75, 90 and 100. The values for other hyper-parameters are weight decay: 0.0035 (0.0007 for WideResNet), momentum: 0.9, and batch size: 128.

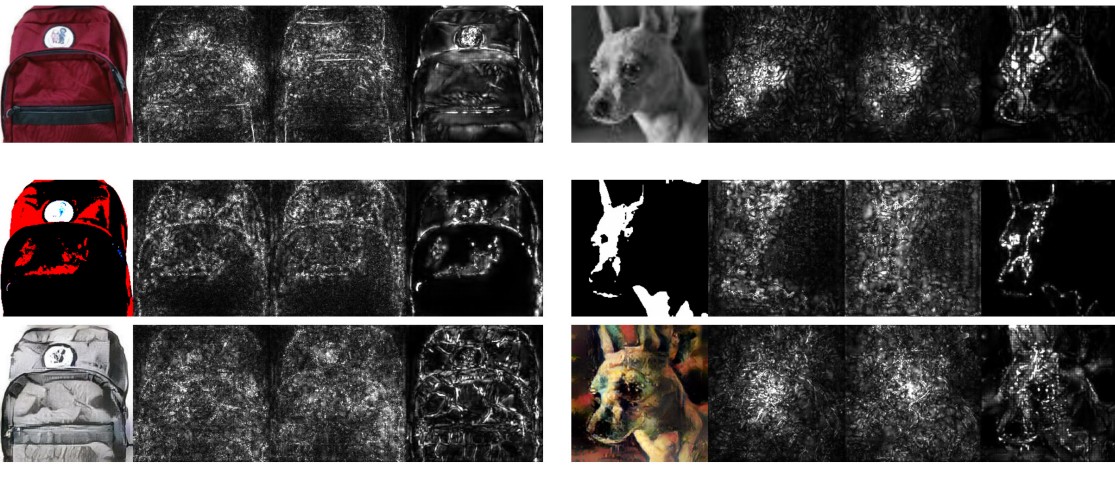

(a) Images from Caltech-256          (b) Images from Tiny ImageNet

Figure 5: Figure 2 from Zhang & Zhu (2019): Sensitivity maps based on SmoothGrad Smilkov et al. (2017) of three models on images under saturation, and stylizing. From top to bottom, Original, Saturation 1024 and Stylizing. For each group of images, from left to right, original image, sensitivity maps of standard CNN, underfitting CNN and PGD-$\ell_\infty$ AT-CNN.

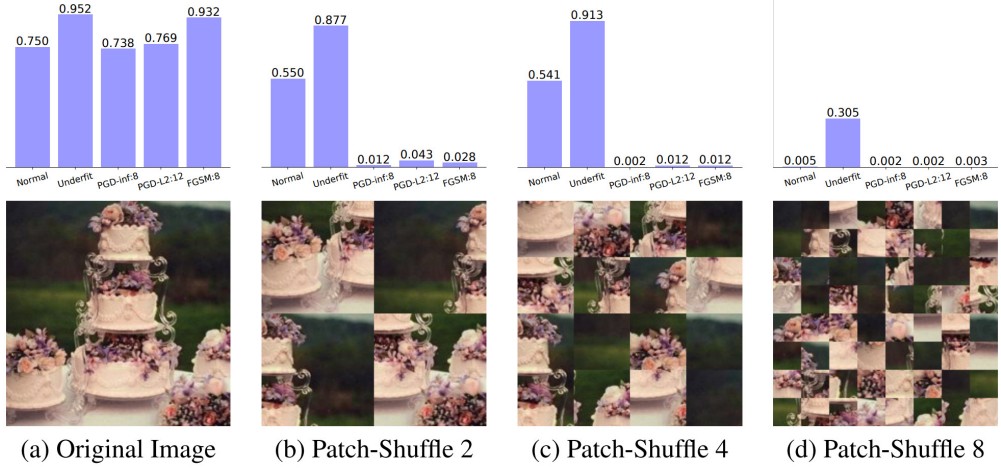

(a) Original Image     (b) Patch-Shuffle 2     (c) Patch-Shuffle 4     (d) Patch-Shuffle 8

Figure 6: Figure 7 from Zhang & Zhu (2019): Visualization of patch-shuffling transformation. The first row shows probability of "cake" assigned by different models.

| Dataset | *strongM/weakM* | Model Architecture | Model Source |
|---|---|---|---|
| CIFAR-10 | *weakM* | WideResNet-28-10 | Standard* |
| | *strongM*-SAT | | Trained locally |
| | *strongM*-TRADES | | Trained locally |
| | *strongM*-AWP | | Wu2020Adversarial* |
| | *strongM*-AWP_E | | Wu2020Adversarial_extra* |
| | *strongM*-WANG23 | | Wang2023Better_WRN-28-10* |
| ImageNet | *weakM* | WideResNet-50 | From PyTorch: wide_resnet50_2 |
| | *strongM*-SAT | | Salman2020Do_50_2* |

Table 12: Source for different WideResNet models. * indicates that the models are obtained from Robust-Bench Croce et al. (2021) and the corresponding source column contains the Model-ID

# E   Parameters used for SQA attacks

| Method | Hyperparameter | CIFAR-10 | ImageNet |
|---|---|---|---|
| Square | p (fraction of pixels changed every iteration) | 0.05 | 0.3 |
| SignHunter | $\delta$ (finite difference probe) | 8([0, 255]) | 0.1([0, 1]) |
| SimBA | $d$ (dimensionality of 2D frequency space) | 32 | 32 |
| | $order$ (order of coordinate selection) | random | random |
| | $\epsilon$ (step size per iteration) | $\frac{8}{255}$ | $\frac{4}{255}$ |
| NES | $\delta$ (finite difference probe) | 2.55 | 0.1 |
| | $\eta$ (image $l_p$ learning rate) | 2 | 0.002 |
| | $q$ (# finite difference estimations / step) | 20 | 100 |
| Bandit | $\delta$ (finite difference probe) | 2.55 | 0.1 |
| | $\eta$ (image $l_p$ learning rate) | 2.55 | 0.01 |
| | $\tau$ (online convex optimization learning rate) | 0.1 | 0.01 |
| | $Tile\ size$ (data-dependent prior) | 20 | 50 |
| | $\zeta$ (bandit exploration) | 0.1 | 0.1 |

Table 13: Hyper-parameters as used for SQA attacks

We used the same parameters as used by AAA defense for most of the attacks. We adapted the code from BlackBoxBench (`https://github.com/adverML/BlackboxBench`), except for SimBA, where we use SimBA-DCT as simple SimBA is unable to attack adpative attacks like ours and AAA. Further we provide the JSON files that have the values of parameters we used for the attacks. The details of the parameters have been compiled in Table 13. For the square attack, for which we used code provided by auto attack, we used p_init = 0.05 for CIFAR-10 and 0.3 for ImageNet.

