# OpenReview forum: "AlignFix: Fixing Adversarial Perturbations by Agreement Checking for Adversarial Robustness against Black-box Attacks"
_TMLR — Accepted by TMLR_

### Review · Reviewer_Bh7h · 2025-06-12

**Summary Of Contributions:**

This paper introduces a novel defense mechanism called Error Correction by Agreement Checking (ECAC), which draws inspiration from human visual perception to enhance the robustness of neural networks against black-box adversarial attacks. The core idea behind ECAC is to leverage the inherent disagreement between naturally trained models (WeakM) and adversarially trained models (StrongM) as a signal for potential errors. When a disagreement occurs, ECAC nudges the input towards the prediction of the WeakM model, effectively correcting adversarial perturbations and expanding the decision boundary of the robust model. The new knowledge presented is that this biologically inspired error correction mechanism significantly improves adversarial robustness across diverse attack types, including score-based, decision-based, and transfer-based black-box attacks, even when the adversary has full access to the model. The authors demonstrate that ECAC consistently outperforms existing baselines on CIFAR-10 and ImageNet datasets and maintains its effectiveness across different architectures like Vision Transformers (ViT). Furthermore, the paper highlights the value of integrating error correction principles into model design, suggesting a new direction for developing more robust AI systems.

**Audience:**

Yes

**Broader Impact Concerns:**

(1) Potential for Misuse in Adversarial Attacks (Offensive Capabilities): While ECAC is designed as a defense mechanism, research into adversarial robustness inherently provides insights into how models can be fooled. The knowledge gained from understanding model vulnerabilities and developing defenses could, in theory, be leveraged by malicious actors to develop more sophisticated and effective adversarial attacks. A Broader Impact Statement should acknowledge this dual-use potential and discuss the responsibility of researchers to mitigate such risks. (2) The paper explicitly states that ECAC incurs a "higher inference time, roughly five times higher, due to the extra forward and backward pass". This significant increase in computational resources could lead to environmental impact and accessibility and equity concerns.

**Claims And Evidence:**

Yes

**Requested Changes:**

Critical for Acceptance:
(1) In-depth Discussion and Mitigation Strategies for Increased Inference Time: The five-fold increase in inference time is a significant practical drawback. While acknowledged as a "practical tradeoff," for acceptance, the authors need to:  (A) Quantify Impact More Precisely,
Propose Concrete Mitigation Strategies and elaborate on specific safety-critical contexts where this increased latency is acceptable.
(2) While Algorithm 2 is provided, its explanation could be improved for broader accessibility.

**Strengths And Weaknesses:**

Strengths:
(1) Novelty and Biological Inspiration: The paper presents a novel defense method, ECAC, inspired by error correction in human visual perception, which is a unique and intriguing approach to adversarial robustness. (2) Focus on Realistic Black-box Attacks: ECAC is specifically designed to mitigate realistic black-box threats, which is a crucial area in adversarial machine learning given that attackers often lack full model access. (3) Effectiveness Across Diverse Attack Types: The experimental results demonstrate ECAC's strong performance against a wide range of black-box attacks, including Score-based Query Attacks (SQA), decision-based, and transfer-based attacks. (4) Validation for Disagreement: The paper provides a clear and experimentally validated rationale for using model disagreement (WeakM and StrongM) as an error signal, highlighting that adversarial examples transfer poorly between these models. (5) The ability to provide comprehensive experiments evaluation across various datasets (CIFAR and ImageNet) and provide detailed results in multiple tables.

Weaknesses:

(1) Increased Inference Time: A significant limitation acknowledged by the authors is the substantial increase in inference time (approximately five times higher) due to the extra forward and backward passes required by ECAC. The authors state this might be "acceptable in safety-critical contexts", but a more in-depth discussion on potential strategies to mitigate this overhead, or specific scenarios where this tradeoff is truly acceptable, would be beneficial.(2) Limited Scope of Robustness: While effective against black-box attacks, the paper notes that ECAC "does not necessarily extend to all types of robustness. This limitation could be further explored, perhaps with a discussion of why these specific transformations are not handled and if there are plans for future work to address them.
(3) Adaptive Attack Algorithm Details: While Algorithm 2 is provided, a more explicit breakdown and explanation of its steps, particularly for readers less familiar with adaptive attack methodologies, could improve clarity.

---

> ### Author Response · Authors · 2025-07-10
> **Response to Reviewer Bh7h Feedback and Revision Summary**
>
> We thank reviewer Bh7h for the prompt and thorough review and appreciate their recognition of our key contributions. Please find our rebuttal and corresponding changes below. We have also updated the manuscript accordingly (changes in blue), and renamed our method from ECAC to AlignFix as per another reviewer’s suggestion.
>
> `>> Increased Inference Time:`
>
> - AlignFix incurs $5\times$ overhead due to one backward and two forward passes per input. A simple ResNet-18 model takes $\approx 1.6$ seconds to classify the entire CIFAR-10 test set, with a batch size of 500 using an RTX-2080 Ti graphics card, whereas AlignFix takes $\approx8.3$ seconds. This is a tradeoff for strong robustness under black-box threats, including decision-based and transfer attacks where prior defenses fail.
>
> - As discussed in Section 3.1, biological perception relies on corrective feedback, which contributes to its robustness. While adversarially robust feed-forward models are ideal, current evidence suggests this may be "fundamentally limited". Despite added cost, AlignFix remains practical e.g., classifying $10K$ CIFAR-10 samples took $8.3s$ on an RTX 2080 Ti. This real-time performance is acceptable in safety-critical settings like medical triage, surveillance, parts of autonomous driving like street sign detection, or offline robust curation.
> - Mitigation strategies include gradient approximations, model distillation, or hardware acceleration.
>
> We added this discussion in the revised Limitations section.
>
> `>> Robustness to Non-Adversarial Transformations: `
> AlignFix is designed for adversarial robustness under realistic black-box $\ell\_\infty$ threats. As shown by Engstrom et al. (2019), adversarial robustness does not automatically imply robustness to spatial transformations. Like other adversarial defenses (e.g., AAA, AT), AlignFix does not address such transformations directly. Handling them requires strongM and weakM to be trained for spatial robustness. This is an orthogonal challenge that can be incorporated into AlignFix by substituting spatially robust base models.
>
> This clarification has been added to the Limitations section.
>
> `>> Elaborating Adaptive Attack algorithm:`
>
> We revised Section 4.2.3 to better motivate the need for adaptive attack. And later we explained the algorithm in detail.

---

### Review · Reviewer_sqwq · 2025-06-19

**Summary Of Contributions:**

This paper proposes a new method towards adversarial perturbations in neural networks, which utilizes the disagreement between a naturally trained network (weakM) and an adversarially trained network (strongM) to perform an error correction for the decision. Specifically, it observes that, when an attack is performed by utilizing both weakM and strongM, then perturbing the example towards the correct class would increase the disagreement between weakM and strongM, and vice versa (observation elaborated in Section 3.2). This information is utilize to nudge the input and reduce the possible errors in making decision. That is, the input will be perturbed towards the class that weakM predicts, and if strongM agrees the class on the perturbed example, the class will be returned; otherwise, the algorithm would trust the decision of strongM. Experiments on multiple attack models validates the effectiveness of the proposed defense against adversarial examples.

**Audience:**

Yes

**Broader Impact Concerns:**

The reviewer does not have extra concern for this part for now.

**Claims And Evidence:**

No

**Requested Changes:**

While the paper claims solving an adversarial attack problem by inspirations from biological observation of human brain (which makes mistake within a very short period of time), the key model observation and true technical contribution seem much more limited than that. To this end, the paper is a bit overclaimed.

The title and abstract (also part of the introduction) need to be rephrased. Especially the abstract includes very limited information about the proposed method, the truly concerned problem, or any insightful observation.

While it seems to claim that the defense is mainly for black-box attacks, the later part shows the defense model is designed for all kinds of attacks. This needs to be clarified.

Section 4 needs to include more quantifiable comparisons of the proposed method and other baseline methods. It would be better if it elaborates on the choice of the baseline (and how it relates to the key observation).

**Strengths And Weaknesses:**

Strengths:

This paper is based on experimental observations, and hence designs an algorithm that utilizes an observed phenomenon (the disagreement between strongM and weakM) to nudge the input and hence corrects possible errors (perturbations). The observed insight is very interesting, and the proposed defense is validated on a variety of attack models and adversarial training models.

Weaknesses:

The concept "error correction" does not quite accurately describe the method, as the algorithm seems not to correct any error, but more to perturb and gather more information. In this sense, the title and abstract part are misleading and less informative. The process of the proposed method and how human brains correct the information is not quite related, and the introduction part somehow implies the method is inspired by biological experiments on how human or primate brains "correct errors in signals". See summary of contribution. This is quite misleading.

On the other hand, I wonder simply claiming that strongM is just some type of adversarially trained model and definitely includes more structure features would be a bit limited. Hence, the paper needs to illustrate more on the key observation. That is, how many adversarially trained models did you use to verify the observation? Does it just exist in some defense types but not the others?

Although the paper already includes lots of adversarially training models, it only presents part of the results. For example, Section 4.2.1 only shows the comparison with AAA, while other comparisons are hard to find.

---

> ### Author Response · Authors · 2025-07-10
> **Response to Reviewer sqwq Feedback and Revision Summary**
>
> We thank reviewer sqwq for the valuable feedback and are glad they found our insight “very interesting.” Please find our response below; all manuscript changes are marked in blue.
>
> `>> Regarding error correction comparison with primate brains, rewording title, abstract, and introduction:`
>
> We agree that biological feedback mechanisms are far more complex. Our use of this analogy is limited to the idea of feedback, not its implementation. To avoid confusion, we have:
>
> a) Renamed our method to AlignFix. Note that when we 'nudge' the input, it can correct adversarial perturbations in some cases, as demonstrated in Section 3.3.
>
> b) Revised the title, abstract, and introduction to better reflect the actual method, and
>
> c) Clearly stated that biological comparisons are only an inspiration.
>
> `>> Regarding the claim that adversarially trained models capture more structural features (shape bias):`
>
> This observation is grounded in the findings of Zhang & Zhu (2019) (referenced in Section 3.2), who conducted systematic qualitative and quantitative comparisons between adversarially trained CNNs (AT-CNNs) and normally trained CNNs. They demonstrated that AT-CNNs exhibit reduced texture bias and increased shape bias. Below is some of their key observation:
>
> * Qualitatively, Figure 2 in their paper (based on SmoothGrad salience maps) shows that AT-CNNs consistently highlight object contours across clean, saturated, and stylized images, whereas standard CNNs produce noisy, texture-focused maps.
>
> * Quantitatively:
>   * Their Table 2 shows that on *stylized images* (which distort textures but preserve shape), AT-CNNs outperform standard CNNs by a wide margin, confirming texture invariance.
>   * Their Figure 7 demonstrates a steep drop in AT-CNN accuracy under *patch-shuffling* (which preserves local textures but destroys shape), while standard CNNs remain largely unaffected, confirming AT-CNNs' reliance on shape.
>
> To clarify this in our paper, we have expanded Section 3.2 and included illustrative figures from Zhang & Zhu (2019) in the Appendix. Our own empirical results (Table 2) further support that this shape bias persists across adversarial training variants, reinforcing that the effect is not limited to a specific defense type.
>
>
> `>> Clarification on black-box vs other attacks.`
>
> Our defense is designed for realistic black-box settings, including adaptive attacks where the adversary knows the defense mechanism but lacks access to exact model weights. Full white-box access is out of scope. To clarify this, we:
>
> a) revised the second contribution to explicitly mention the black-box scope, and
>
> b) also clearly stated this limitation in the 'Limitations' section.
>
> `>> Regarding elaboration of the baseline.`
>
> AAA is our main baseline since it explicitly targets black-box defenses. It compares against undefended, RND, and adversarially trained models. We updated this clarification in Section 2.3 (Related Work). Additionally, in Section 4, we report AlignFix results across a variety of adversarially trained models and demonstrate consistent improvements, covering a broader set of baselines than AAA.

---

### Review · Reviewer_MXpr · 2025-06-24

**Summary Of Contributions:**

This paper proposes a novel adaptive approach to defend classification networks against adversarial attacks. The focus is primarily on black-box attacks, a challenging setting where many existing defense strategies fail. ECAC uses the fact that naturally and adversarially trained models rely on different feature representations. While adversarially trained models are robust by design, naturally trained models often resist attacks targeting their adversarially trained counterparts. ECAC uses both models jointly, adjusting inputs toward the naturally trained model's prediction only when their outputs agree, ensuring reliable error correction without sacrificing robustness. Experimental findings show that the proposed method performs competitively.

**Audience:**

Yes

**Broader Impact Concerns:**

ECAC improves robustness against practical black-box attacks but increases inference time, which may limit accessibility in resource-constrained settings. It also does not cover all types of robustness, leaving residual risks in safety-critical applications.

**Claims And Evidence:**

Yes

**Requested Changes:**

- I would appreciate clarification regarding Figure 1. As I understand it, the ECAC method utilizes the output of either the weak or strong model when both predict the same class. In cases of disagreement, the final prediction is taken from the strong model evaluated on the clean, unperturbed input. If this interpretation is correct, it raises a concern: relying on the clean image to improve robustness does not constitute an actual defense mechanism. As long as the clean input x is used instead of the adversarially perturbed input x′, the system is not exposed to an attack in the first place.
Alternatively, if x in this context represents the perturbed input, could you clarify whether ECAC actively modifies or corrects this perturbed input before passing it to the strong model? This distinction is critical, as the current presentation introduces ambiguity.
This confusion is further exacerbated by the inconsistent use of notation: in Sections 2.1 and 2.2, the adversarially perturbed image is clearly denoted as x′. If, in Figure 1 or elsewhere, x′ is also being used to represent inputs perturbed for defense purposes, I would strongly recommend adopting distinct symbols to differentiate between adversarial perturbations and any modifications applied as part of the defense. Clear and consistent notation would greatly improve the interpretability of the method.

- The details of the validation experiments are not provided in the paper. The only reference appears on Page 7, where it is stated that the first 2,000 samples are used as the validation set. A comprehensive explanation of the hyperparameter tuning process and validation experiments, at least for one representative experimental setup, is necessary to clarify the methodological details. Additionally, it is unclear whether these 2,000 validation samples were selected randomly. Please clarify this point.

- Regarding Table 3, it is mentioned that "AAA-Linear evaluated Square Attack on 1,000 randomly selected samples (one per class) that were correctly classified by the naturally trained model." However, the test setting for ECAC is not clearly described. Were the same 1,000 samples used to evaluate ECAC, or were the test images for ECAC selected independently? Please provide clarification.

- Table 3 reports the defense performance on the large-scale ImageNet dataset. On this dataset, AAA-Linear outperforms ECAC for several black-box attacks, including SignHunter, NES, and Bandit, raising concerns regarding the relative effectiveness of the proposed method. Including additional results, such as evaluations on other datasets like CIFAR-100, would offer a more comprehensive perspective and facilitate a fairer assessment of ECAC's performance.

- Revise the paper to correct typographical and notation errors, such as the use of s_size in Section 3.4.

**Strengths And Weaknesses:**

Strengths:

- The work addresses a critical and timely issue within its research domain.
- The motivations and objectives are clearly articulated and well justified.
- ECAC offers a simple yet highly effective defense against a broad range of black-box attacks, including score-based, decision-based, and transfer-based methods.


Weaknesses:
- The paper is difficult to follow at the beginning, as it starts with method comparisons without proper context or citations (see Table 1, Page 1). It would be clearer to either move the table after the relevant explanation or bring forward the accompanying discussion to better support and clarify the table content.
- The explanation of Figure 2 lacks clarity. It is not clear whether P_y​ refers to the predictions of StrongM, WeakM, or the ECAC method. Additionally, the meaning of the colors used in the figure is ambiguous. Does blue indicate correct classifications and pink incorrect ones, or vice versa? Please provide a more detailed explanation to clarify the figure, including what each symbol and color represents, and how it relates to the methods discussed.
- See the requested changes for more details.

---

> ### Author Response · Authors · 2025-07-10
> **Response to Reviewer MXpr Feedback and Revision Summary**
>
> We thank reviewer MXpr for the feedback. We have addressed the concerns, updated the manuscript accordingly (changes in blue), and renamed our method from ECAC to AlignFix as per another reviewer’s suggestion.
>
> `>> Clarity issue in the beginning; Table 1 presented without sufficient context.`
>
> We revised the introduction to clarify the key motivation and added the necessary context before Table 1. The table now appears after a discussion of model disagreement and robustness features, making it easier to follow. We also added a concise summary of our method in the abstract for early clarity.
>
>
> `>> Clarifying Fig-2: ambiguity in P_y, color, etc.`
>
> We updated the caption of Figure 2 to specify that P_y refers to the correct class logit and clarified the meaning of all symbols. To avoid confusion with Figures 1 and 3, we changed the color scheme to green for correct class logits and red for incorrect ones.
>
> `>> Regarding clarity of notation in Figure 1 and interpretation of the defense mechanism:`
>
> We clarified in the paper that input $x$ in Figure 1 refers to the actual input received at inference time, which may be clean or adversarially perturbed. To eliminate ambiguity, we now denote the AlignFix-nudged input as $\tilde{x}$, clearly distinguishing it from $x'$, which refers to adversarial inputs crafted during attacks. This distinction has been applied consistently throughout the paper and figure.
>
> AlignFix does not assume access to clean inputs. It nudges the received input toward WeakM's prediction and returns StrongM($\tilde{x}$) if agreement is achieved, or StrongM($x$) otherwise.
>
> `>> Regarding hyperparameter tuning:`
>
> We revised the last paragraph of Section 3.4 to clarify our tuning process. For CIFAR-10, we performed ablations (Appendix B, Table 9) to select the step size, with n_steps fixed at 1. The effective hyperparameter is the nudging magnitude (s_size × n_steps), and s_size = 0.02 was selected heuristically, based on the intuition that nudging should be around half the perturbation budget (ϵ = 0.031). As shown in the ablation study (Appendix B, Table 9), increasing s_size improves robustness to SQA attacks but reduces transfer robustness, and 0.02 provides a good tradeoff between the two.
>
> For ImageNet, the 2K validation samples used for testing were selected by sorting image filenames alphabetically and picking the first 2K. This detail has been added to the text. The resulting sampling is consistent yet effectively random, as class labels are randomly distributed with respect to filename order.
>
> `>> Regarding test set for ECAC (AlignFix) on ImageNet`
>
> We clarified the evaluation protocol in the text. The AAA paper reported results on 1,000 samples correctly classified by the naturally trained model and scaled accuracy by 78.48% (natural accuracy). In contrast, we report the results, including re-evaluation of our baseline AAA, on the first 2,000 validation samples without filtering. This consistent, unfiltered setting slightly favors AAA, yet AlignFix remains competitive and outperforms it on Square and SimBA attacks.
>
> `>> Regarding AAA outperforming ECAC (AlignFix) on certain attacks:`
>
> While AAA outperforms AlignFix on some attacks, AlignFix provides better worst-case robustness, which is more relevant under realistic threat models. A practical adversary would favor the most damaging attack, like Square attack, where AlignFix consistently outperforms AAA. This evaluation aligns with AAA's setup and the datasets used, as it holds across both CIFAR-10 and ImageNet. While CIFAR-100 evaluations are technically feasible using AAA’s released code, we focused on CIFAR-10 and ImageNet to reflect the datasets emphasized in AAA’s original evaluation.
>
> `>>` We corrected the typographical and notation issues in Section 3.4, including the use of s_size.

---

### Decision · Action_Editor_Mjnb · 2025-08-12

**Recommendation:** Accept with minor revision

**Audience:**

Yes

**Audience Explanation:**

This paper proposes AlignFix, a defense to improve the adversarial robustness of a classifer, based on agreement checking techniques. The method is shown to be particularly useful for black-box attacks. While the reviewers are mostly satisfied with the revision, there are some mandatory changes to the claims that need to be made. Some reviewers also pointed out the limitation of increased inference cost and limited performance gain compared to recent defenses such as AAA. However, this is a limitation rather than a technical flaw.

Suggested Changes:
- Claim on robustness to white-box attacks: The authors state that "**While it retains significant robustness even under white-box attacks, this is out of scope for the current work.**" However, the focus of this paper is on defense against black-box attacks. Since there is insufficient evidence on the robustness against white-box attacks, the statement is not grounded and should be revised.

**Claims And Evidence:**

No

**Claims Explanation:**

Most claims are supported by empirical results. The claim on white-box attack needs to be updated.

---

> ### Author Response · Authors · 2025-08-15
> **Revision addressing white-box robustness claim per final decision**
>
> We thank the reviewers for the final acceptance. We have addressed the raised concern as follows:
>
> 1. The abstract no longer includes the claim about robustness when the adversary has full access to the model.
> 2. The original statement has been revised to:
> "The AlignFix defense, by always outputting the \textit{strongM} logit, theoretically constrains the total perturbation to the sum of the adversarial attack's perturbation ($\epsilon$) and the defense's corrective nudge ($step\\_size$). This may provide a degree of white-box robustness, however comprehensive empirical evaluation against white-box attacks is outside the scope of this paper. "
>
> We will submit the camera-ready version shortly.